# Public parks and the pandemic: How park usage has been affected by COVID-19 policies

Zoe M. Volenec[1]◉, Joel O. Abraham[1]◉*, Alexander D. Becker[1,2], Andy P. Dobson[1]

1 Department of Ecology and Evolutionary Biology, Princeton University, Princeton, NJ, United States of America, 2 Department of Biology, Stanford University, Stanford, CA, United States of America

◉ These authors contributed equally to this work.
* joeloa@princeton.edu

**Data Availability Statement:** Data are archived at Dryad Data Repository, under the following DOI: 10.5061/dryad.51c59zw87.

**Funding:** The authors received no specific funding for this work. Publication of this study was

## Abstract

Public parks serve an important societal function as recreational spaces for diverse communities of people, with well documented physical and mental health benefits. As such, parks may be crucial for how people have handled effects of the COVID-19 pandemic, particularly the increasingly limited recreational opportunities, widespread financial uncertainty, and consequent heightened anxiety. Despite the documented benefits of parks, however, many states have instituted park shutdown orders due to fears that public parks could facilitate SARS-CoV-2 transmission. Here we use geotagged social media data from state, county, and local parks throughout New Jersey to examine whether park visitation increased when the COVID-19 pandemic began and whether park shutdown orders were effective at deterring park usage. We compare park usage during four discrete stages of spring 2020: (1) before the pandemic began, (2) during the beginning of the pandemic, (3) during the New Jersey governor's state-wide park shutdown order, and (4) following the lifting of the shutdown. We find that park visitation increased by 63.4% with the onset of the pandemic. The subsequent park shutdown order caused visitation in closed parks to decline by 76.1% while parks that remained open continued to experience elevated visitation levels. Visitation then returned to elevated pre-shutdown levels when closed parks were allowed to reopen. Altogether, our results indicate that parks continue to provide crucial services to society, particularly in stressful times when opportunities for recreation are limited. Furthermore, our results suggest that policies targeting human behavior can be effective and are largely reversible. As such, we should continue to invest in public parks and to explore the role of parks in managing public health and psychological well-being.

## Introduction

The COVID-19 pandemic has disrupted many aspects of human social life, limiting recreational opportunities and affecting peoples' financial security [1, 2]. These disruptions have resulted in profound increases in stress, anxiety, and depression [3–5]. As a result, people have had to seek out alternative means of recreation, both for themselves and for their children, and of managing their mental and physical health [3, 6–8].

supported by the Princeton University Library Open Access Fund.

**Competing interests:** The authors have declared that no competing interests exist.

Public parks have always served a critical function as free recreational spaces [9–12], particularly for low income and immigrant communities [13–16]. Parks and green spaces more generally are known to have positive impacts on mental health [9, 17], with demonstrated mental health benefits [18–20]. Also, parks can be quite expansive, allowing for safe enjoyment while maintaining social distancing (1). As such, parks may serve an important–and as of yet unquantified–function for people in coping with the COVID-19 pandemic [6, 21–23].

The possibly increased appeal of parks made them the focus of policy attention in the United States during the onset of the COVID-19 pandemic [24–26]. Because of their function as recreational spaces for diverse communities, policymakers worried that parks could serve as transmission hotspots for SARS-CoV-2 [1, 6, 23, 25]. Parks, especially those tailored towards children, have many shared surfaces, which has been hypothesized to result in increased disease transmission [1, 24]. Furthermore, parks are often unsupervised and lack guidelines for visitor capacity, making it difficult to regulate their usage [1, 24].

In response to these concerns, some states instituted park shutdowns, entirely preventing people from using these open spaces [25, 26]. Park shutdowns have been criticized somewhat for cutting people off from the benefits of publicly available outdoor areas [1, 6], disproportionately impacting lower income individuals with limited access to green spaces [13, 16, 27, 28]. That said, it is unclear how effective these park shutdown orders were at reducing public park usage and achieving their purpose of diminishing transmission risk. On one hand executive orders shutting down parks may indeed act as a deterrent, but alternatively people may knowingly or unknowingly ignore the park shutdown orders. Attempts to discern the effects of these park closure policies have been limited, hindering the ability of policy makers to effectively evaluate their implementation [6, 21, 26].

Determining the efficacy of park shutdown orders requires case studies of states that have instituted such shutdowns. New Jersey was at the forefront of the United States SARS-CoV-2 outbreak, experiencing large outbreaks relatively early in the pandemic [25, 26, 29]. The state had to lead the charge in crafting and instituting policies to respond to the pandemic, including an executive order by New Jersey's governor Governor Murphy shutting down state and county parks [25, 26, 30]. This executive park shutdown order did not go into effect until nearly a month after stay-at-home restrictions were put in place and was subsequently lifted about a month later (Fig 1) [30]. Thus, New Jersey provides an important case study for understanding both (1) the immediate impacts of COVID-19 on the use of public parks and (2) the efficacy and reversibility (*i.e.*, the ability to return to pre-order conditions) of policies like the park shutdown order.

Here we sought to understand how park usage changed during the onset of the COVID-19 pandemic with a dynamic recreational and administrative landscape. We evaluate three questions. First, how did park usage in New Jersey change with the onset of the COVID-19 pandemic? Second, how did the New Jersey governor's executive order to shut down parks affect the use of public parks? And finally, how did the lifting of the park shutdown order impact subsequent park visitation? We predict that (1) park usage increased significantly with the start of the pandemic, given the crucial role that parks play as widely accessible recreational spaces; (2) the executive order then significantly decreased park visitation to closed parks, but parks that remained open saw higher visitation as people congregated in the remaining open parks; and (3) the withdrawal of the executive order caused park visitation to increase again, possibly exceeding visitation before the closures as people compensated for the lack of access to parks (Fig 1).

## Methods and materials

In this study, we assess visitation in parks throughout New Jersey using geotagged Instagram photography data from February 2017 through May 2020. With these data we

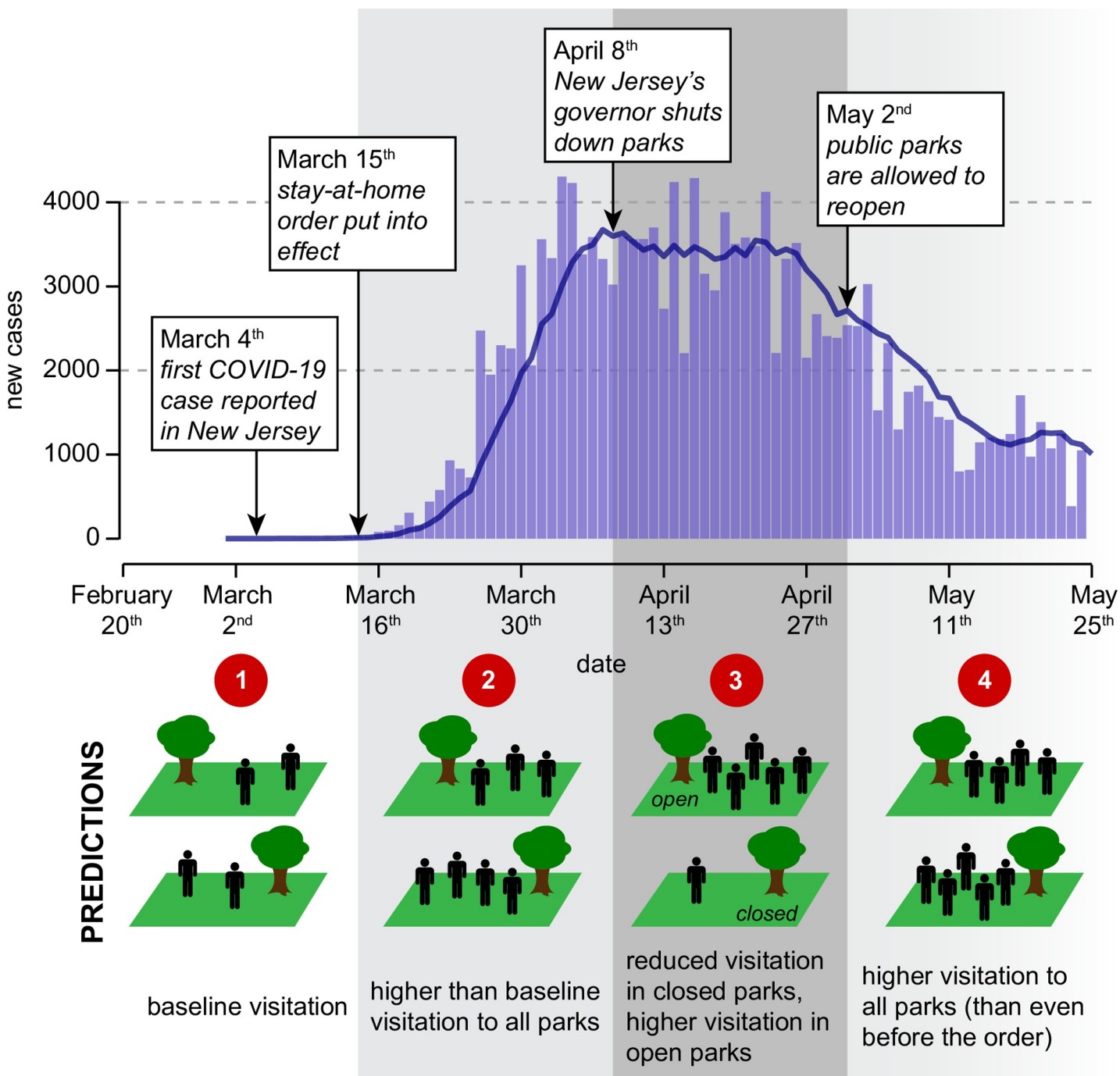

**Fig 1. Daily new COVID-19 cases reported in New Jersey, with key policy events marked and their predicted impacts on the use of public parks depicted.** Light purple bars correspond to the number of new cases per day, whereas the dark blue line reflects the number of new cases averaged over the previous seven days. We predict that the stay-at-home order (period 2) caused people to use parks more (relative to baseline usage before quarantine; period 1). We predict that the subsequent park shutdown order (period 3) was largely effective at reducing park usage in closed parks but possibly concentrated people in the parks that remained open. Finally, we predict that the end of the shutdown order (period 4) caused people to return to parks at higher levels than they had before the shutdown. New Jersey COVID-19 case data are from [31].

compare park usage during four periods of spring 2020: before the pandemic began (period 1); during the beginning of the pandemic (period 2), after the stay-at-home order was instituted but before the park shutdown; during the governor's park shutdown order (period 3);

and following the lifting of the shutdown (period 4), when parks were allowed to reopen (Fig 1).

## Study area

New Jersey is the United States' most densely populated state, but, despite this, over 6,070 km$^2$ of New Jersey are preserved as open space or farmland, totaling approximately 34% of the state's land area [32]. Our study analyzed the visitation patterns in 98 parks within this substantial park network (Fig 2). Parks were located in the northern and central portions of the state and span the Piedmont Plains and Skylands ecoregions, which together stretch from the Appalachian Mountains to the Atlantic Ocean and encompass a range of habitats [33, 34]. Though once widely cultivated, these regions are now largely dominated by successional pine and hardwood woodlands [34, 35]. However, the Piedmont Plains and Skylands ecoregions have been subject to increasing conversion to urban and suburban land cover as a result of New Jersey's population growth over the past several decades [34, 36].

We used the New Jersey Open Space and Preservation Resource Inventory (NJ OSPRI) to identify candidate parks for inclusion in this study (Fig 2) [37]. We defined parks broadly, in line with the categories included in the NJ OSPRI dataset, and therefore included conservation areas, preserves, recreational facilities (*e.g.*, athletic fields), conservation easements, and historic sites. Altogether, we identified 13 state parks, 5 wildlife management areas, 14 county parks, and 66 local parks for inclusion in this study. As such, the parks we selected capture a broad range of recreational uses and fall under a variety of municipality and agency management categories.

## Park visitation data

Previous work has demonstrated that social media data can be used to approximate park visitation rates, as the two are highly correlated [38–40]. We therefore assessed park visitation at our selection of candidate parks using publicly available, geotagged social media data from Instagram [38, 41]. Instagram is an increasingly widely used online social media platform in which users can upload and share photos with their friends or 'followers' [41–43]. Alongside photos, Instagram users can provide captions and choose georeferenced location tags from a list of potential geotags within the application. Users can make their accounts either private, such that only the user's followers can see their account and posts, or public, such that anyone can see their pictures as well as photo captions and tags.

To gather data on park usage, we wrote a Python script to collect data from publicly accessible Instagram photos (in accordance with Instagram's Terms of Use) using the Selenium WebDriver package [44]. We first matched the parks identified above to their corresponding Instagram location tags, of which there were often several for a park, and visually verified that all location tags corresponded to the correct parks using Google maps. After we had created this list of accurate location tags for the parks that we identified in the NJ OSPRI dataset, we ran our Python script to collect the metadata of all photos associated with each tag; we collected the user handle, date, location tag, photo caption, and hashtags for each photo. We gathered data on photos posted from 2017 to present, as the high volume of photos from some popular parks prohibited the feasibility of collecting pre-2017 data.

After we compiled data on all the photos tagged at these 98 parks, we verified that no photos from parks in other locations were accidentally included in our dataset. This was necessary because parks with shared names but located in other states were occasionally mistakenly included on our park location pages (*e.g.*, photos from 'Warren County, Georgia' ended up on 'Warren County Park, NJ' location page). We also aggregated data for parks that had several

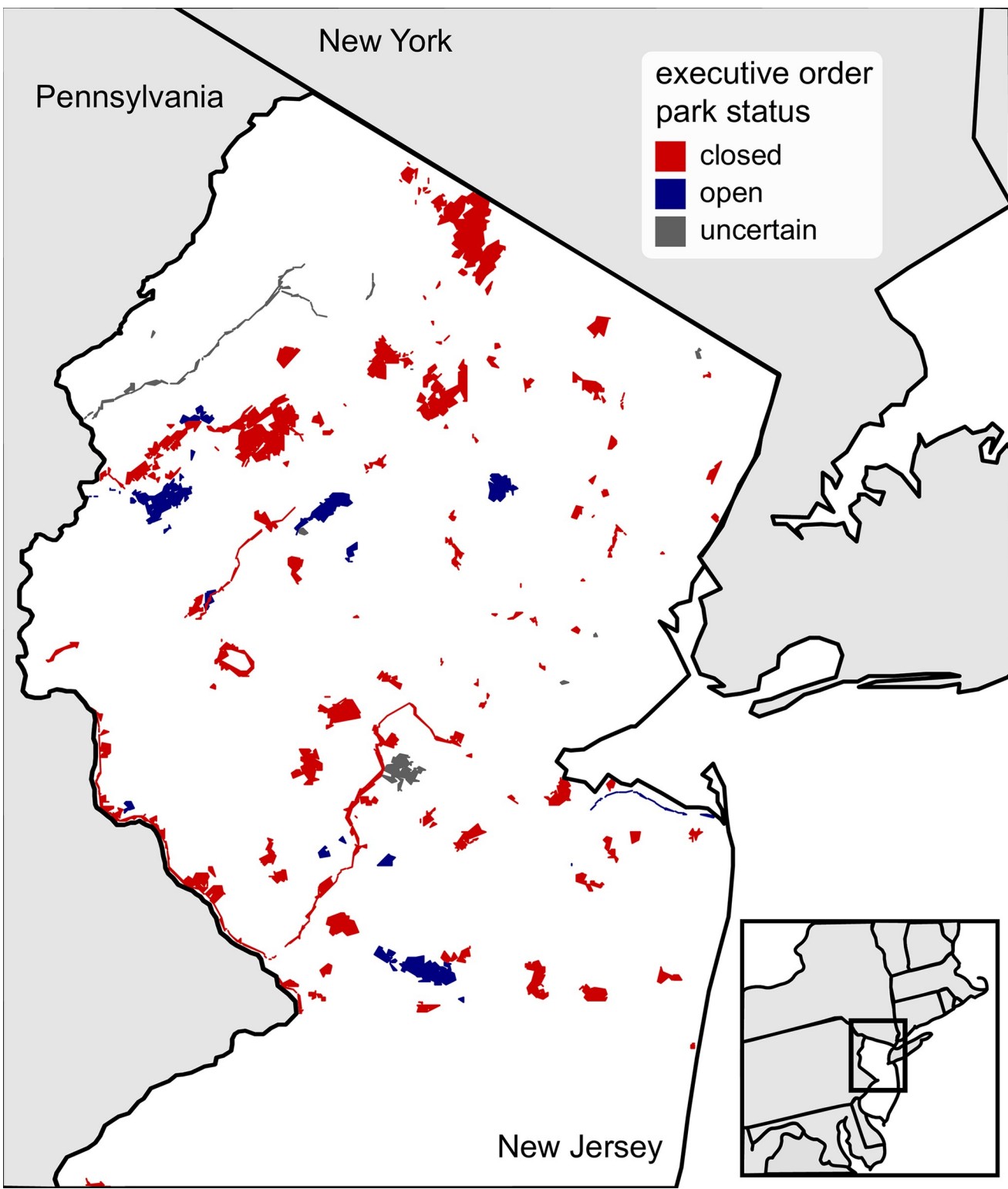

**Fig 2. The distribution of New Jersey parks included in this study (*n* = 98).** Parks in blue (*n* = 12) remained open through the whole study period, whereas parks in red (*n* = 79) were closed during the governor's executive order (April 8th—May 2nd). For parks colored in grey (*n* = 7), we were unable to find information on their status during the park shutdown order, so these parks were excluded from all analyses comparing closed and open parks.

different location tags corresponding to different permutations of the park's name (*e.g.*, photos tagged 'Delaware and Raritan State Park' and 'D&R Canal State Park' were taken to represent the same location).

Once photos were assigned to the correct parks, we performed two final cleaning steps on our dataset: (1) we removed all photos with 'tbt', 'throwback', and 'flashback' in the tags or comments section, because these are not necessarily reflective of park visitation during the study period; and (2) we subsetted our dataset for parks with data from at least every other week in 2020 ($\geq$11 weeks), to ensure adequate data coverage for drawing comparisons across parks. Also, due to particular concerns that photos posted during the park shutdown might have been retrospective posts, we went back through photos from the closure period and manually inspected photo comments and tags to remove those that made obvious reference to park trips from before the governor's order.

There were a few limitations to this approach. First, we were only able to use geotagged photos from public accounts using this method, though we can see no obvious reasons this would bias our findings; there is no reason to expect that people with private accounts would post preferentially at specific parks or on certain days. Secondly, Instagram has a feature called 'Instagram stories' in which users can post photos temporarily (24 hours) for their followers to see [45]. Instagram stories have increased in popularity since the feature's introduction in 2016 [45]. As stories are transient and not public, our method could not capture data from Instagram stories. However, if users increasingly created stories rather than making posts, this would lead us to underestimate park visitation in 2020, thereby underestimating any differences we find between 2020 and previous years. As such, we feel our data are able to accurately capture changes in relative park visitation, both within and across years [38–41].

Altogether, this methodology resulted in a dataset of 160,079 Instagram photos from 98 parks, spanning 2017 to 2020. Of these, 24,807 (15.5%) were from the first five months of 2020 alone.

## Other model variables

The beginning of the COVID-19 pandemic was contemporaneous with the onset of spring and resultant changes in weather [25, 26, 29]. Since the use of public parks and outdoor recreational spaces is highly dependent on the weather, increasing with warmer temperatures and decreasing with precipitation [46], we were concerned that the changes in weather associated with the onset of spring might drive our results (S2A Fig). For instance, park usage would be expected to increase during quarantine simply because of concomitant increases in temperature [46]. To account for this, we gathered daily temperature and precipitation data for Ewing Township, NJ, which is located in central New Jersey and therefore broadly representative of weather conditions across all parks [47]. We used weather data to detrend visitation data and compare park usage across time periods when applicable (see *Statistical analyses*). It would have been methodologically impractical to use weather data for each individual park, as we often aggregated data across parks in our analyses (see *Statistical analyses*).

As previously stated, Instagram has increased in popularity over time (S2A Fig) [41–43]. As such, changes in the total number of users–and therefore total number of pictures posted–might drive any observed differences in park visitation between years. To account for this, we compiled data at the finest temporal scale possible–yearly data–on the total number of United States Instagram users [42, 43], so we could detrend visitation data and compare park usage across years (see *Statistical analyses*).

Critical to this study was an accurate list of parks that were closed due to the shutdown order, so we could compare trends between closed and open parks. Though the New Jersey

governor's order to shut down parks applied primarily to state and county parks [25, 26], some other park management agencies followed suit and likewise shut down their parks [25, 26]. As such, we ascertained which parks were closed and which ones remained open from a variety of sources (see complete list of sources consulted in S1 File). Note that for seven out of the 98 parks included here, information on closure status during the executive order was not available (Fig 2). As such, these seven parks were excluded from all analyses that compared closed and open parks.

## Statistical analyses

All data analyses were done in R 3.6.1 [48]. For statistical analyses, we calculated the photograph user days (PUD) as our response variable. PUD is the number of unique combinations of Instagram users and date information for each park [38–40]. Likewise, for analyses in which we were aggregating across other time scales, not days, we used the number of unique Instagram users for that time period. In both cases we did this to avoid inflation in our visitation metrics from individuals who uploaded several photos of the same park on the same day and to prevent any photos that may have been included under several different permutations of a park's location tag from being double (or triple) counted.

**Role of public parks in the pandemic.** To assess whether the onset of the pandemic and the stay-at-home order were associated with increased park visitation, we compared park usage during the first month of the pandemic (period 2) to park visitation during the same time period in previous years (with the date range adjusted slightly to include the same number of weekdays and weekends). We first detrended PUD data using annual estimates of the number of active US Instagram users to account for increased popularity of the platform over time (S2 Fig) [41–43]. To accommodate the non-normality of the data and the negative non-integer detrended PUD values, we calculated bootstrapped estimates of model coefficients for all our analyses. For this research question, we modeled detrended PUD, aggregated across parks, as a function of year, maximum daily temperature, and daily precipitation rate (Fig 3A). We considered a model variable to have an effect on park visitation (*i.e.*, 'be significant') if the 95% confidence interval did not overlap zero.

To further determine whether the work from home order was associated with increased park visitation, we compared PUD during the first month of quarantine (period 2) to the equivalent time frame before COVID-19 restrictions were put in place in 2020 (period 1). For this analysis, we did not accommodate the change in Instagram user numbers, as Instagram user data were only available at annual scale. We used our bootstrap methodology to model PUD, aggregated across parks, as a function of pre- and post-quarantine status, maximum daily temperature, and daily precipitation rate (Fig 3B).

**Efficacy of the executive order.** To determine whether the New Jersey governor's park closure order was associated with changes in park usage, we compared visitation patterns before the shutdown (period 2) and during the shutdown (period 3) in parks that were made to close and parks that remained open (Fig 4). However, closed parks vastly outnumbered open parks ($n_{closed}$ = 79, $n_{open}$ = 12). To account for this, we simulated 1,000 datasets with equivalent numbers of open and closed parks ($n$ = 12) and calculated bootstrap estimates of the effect of park closure for each dataset. Furthermore, the parks that closed had substantially higher historical visitation rates than parks that remained open, as they were the more widely used county and state parks (S2B Fig). Therefore, we evaluated the percent change in visitation levels from before and after the executive order, rather than absolute visitation, and how this related to park status (open vs. closed). Temperature was not included as a variable in this analysis since it did not differ between the two time periods (periods 2 and 3) according to a separate bootstrap analysis.

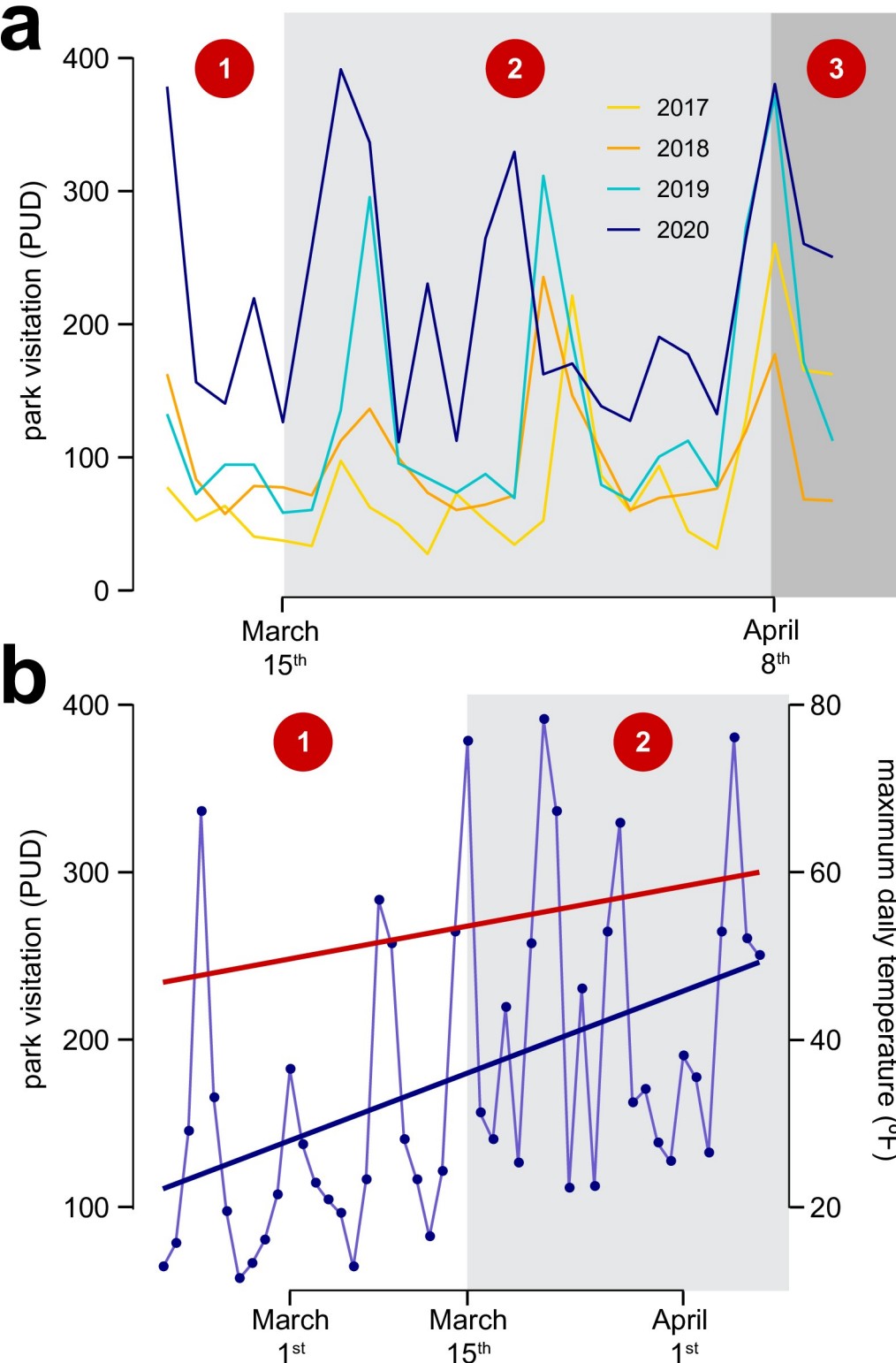

**Fig 3. The effects of the onset of the COVID-19 pandemic on park visitation (as measured by PUD, photograph user days).** (a) Park usage during mid-March through early April of 2020 was higher relative to previous years, and (b) park visitation increased when the stay-at-home order was put into effect, even after controlling for temperature increases. The red trendline in (b) corresponds to temperature trends during this interval, whereas the blue trendline corresponds to the overall trend in park visitation. The numbers in red circles refer to the time periods denoted in Fig 1.

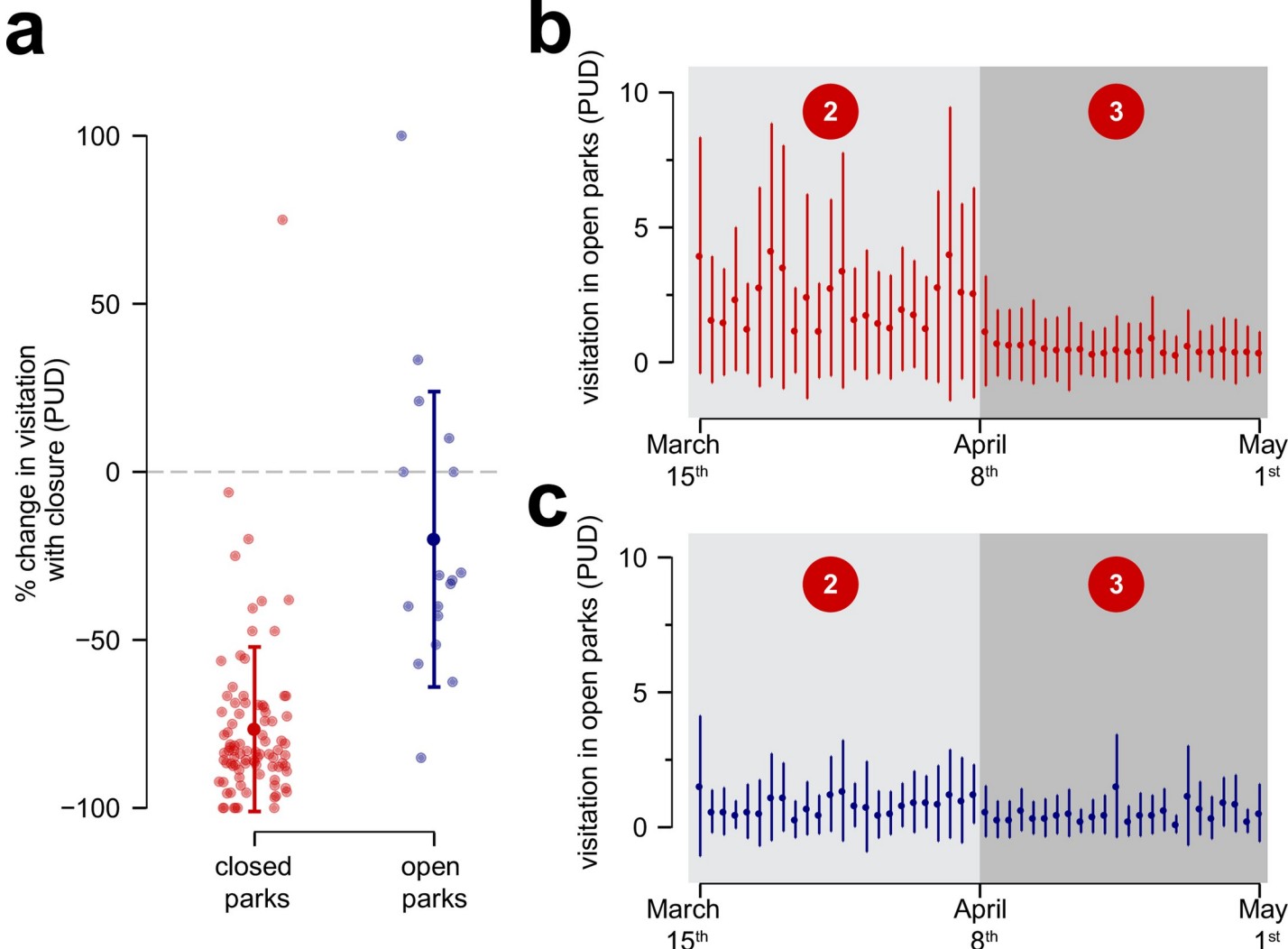

**Fig 4.** (a) The impact of the New Jersey governor's executive order to shut down parks on park visitation (a). (b) Visitation in closed parks dropped substantially, whereas (c) changes in visitation to parks that remained open were much more variable. The numbers in red circles in (b) and (c) refer to time periods denoted in Fig 1.

**Lifting of the executive order.** To determine whether reopening parks was associated with changes in park visitation patterns, we first compared park usage during May 2020, following the lifting of the executive order (period 4), to visitation during the same interval in previous years (Fig 5A). We calculated bootstrap estimates of the effect of year, maximum temperature, and daily precipitation on detrended PUD, aggregated across parks.

Next, we determined how visitation changed in parks after lifting the park closure order in 2020 (comparing periods 3 and 4). We again wanted to assess whether patterns differed between formerly closed parks and parks that had remained open, so we simulated 1000 datasets with equivalent numbers of closed and open parks. For this analysis we detrended PUD by temperature, as temperatures in March/early April differed from those in May. Then we generated our bootstrap estimates of executive order status (during vs. after park closure) and its interaction with park status (open vs. closed) on our detrended PUD measure (Fig 5B).

Finally, we wanted to determine how post-order visitation levels (period 4) compared to park visitation during the initial stages of the pandemic (period 2), to see whether park usage

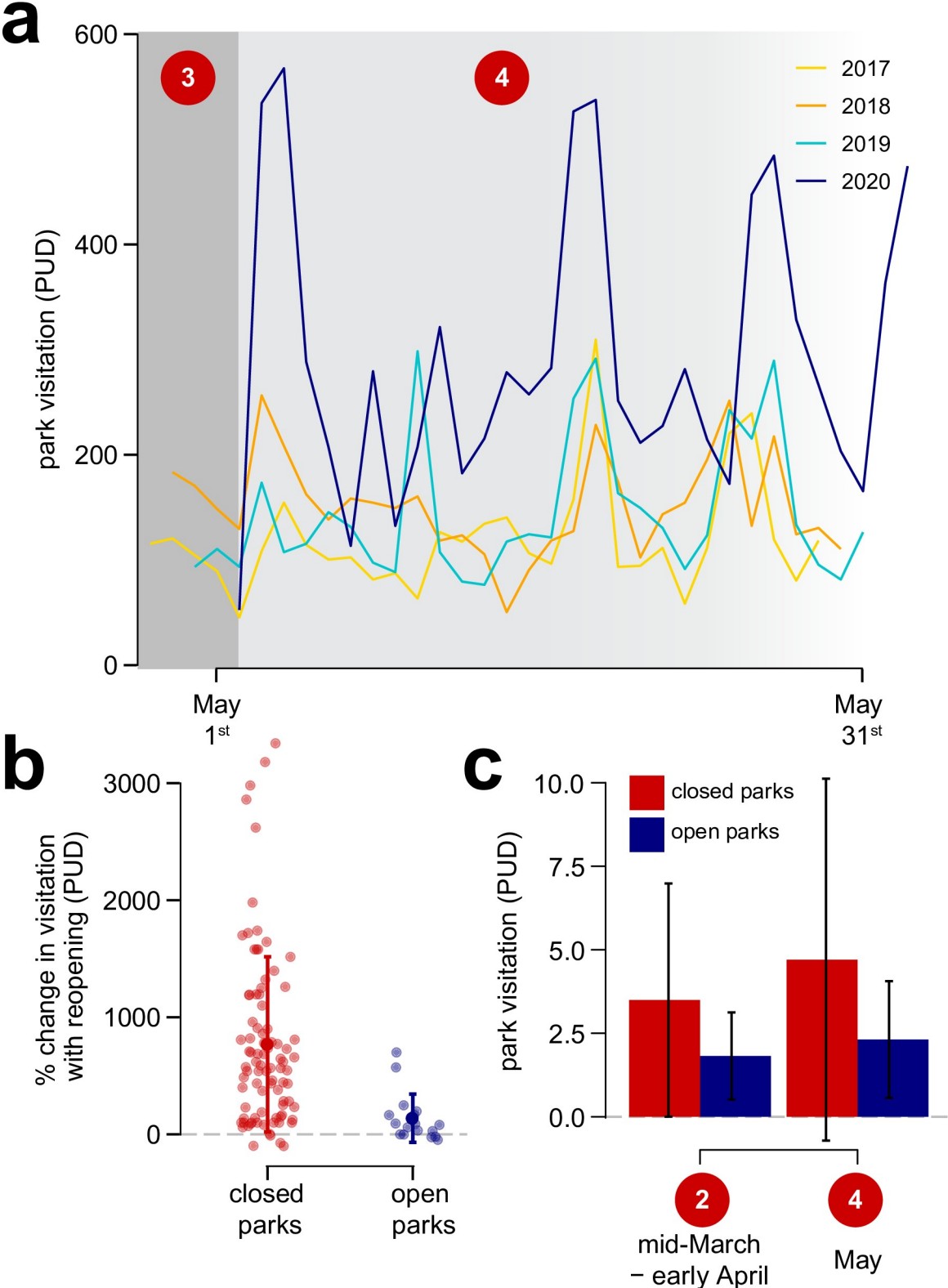

**Fig 5. The effects of the lifting of the park shutdown order on park visitation.** (a) Park usage during May 2020 was higher relative to previous years. (b) Increases in visitation were much larger in formerly closed parks as compared to parks that had remained open. However, (c) visitation did not exceed visitation levels from before the shutdown. The numbers in red circles in (a) and (c) refer to time periods denoted in Fig 1.

increased in the aftermath of the park shutdown, particularly in the parks that had been closed. Again, we detrended PUD by temperature (since temperature differed between the two periods). For this analysis, we subsetted our post-executive order dataset (period 4) to only the first four weeks of May to have an equal number of days and weekends between the two periods. Then we generated our bootstrap estimates of executive order status (before vs. after park closures) and its interaction with park status (open vs. closed) on our detrended PUD measure (Fig 5C).

## Results

### Role of public parks in the pandemic

We found that, as predicted, the onset of the pandemic was associated with increases in park usage. Park visitation in 2020 was higher during the first month of quarantine than during the same time frame in previous years, even when accounting for increased Instagram usage over time and variations in weather (Fig 3A). Likewise, we observed that park visitation was 63.4% higher in the ~3.5 weeks following general COVID-19 quarantine restrictions (period 2, before park closures) than in the preceding 3.5 weeks of 2020 (period 1, before the pandemic), exceeding rates of temperature increase (Fig 3B); the mean daily PUD across parks in period 1 was 123 (+/- 14), whereas mean PUD in period 2 was 201 (+/- 17). Together, these findings demonstrate that the onset of the pandemic was associated with increased utilization of public parks.

### Efficacy of the executive order

We found that the executive order to shut down parks was associated with decreased visitation to state and county parks, suggesting that the shutdown order was largely effective at restricting park usage. While closed parks saw significant declines of 76.1% (+/- 2.9%) in visitation from pre-order levels (Fig 4B), not all parks that remained open saw predicted increases in visitation from pre-order levels, and some even saw slight declines (Fig 4A). In fact, only four of the 12 open parks analyzed here saw increases in visitation following the executive order closing many parks (S1 Fig), which contrasts notably with our prediction that people would concentrate in remaining open parks (Fig 1). On the whole, parks that remained open after the executive order experienced less of a decline, if any, in visitation relative to those that were closed during the shutdown (Fig 4A).

### Lifting of the executive order

Finally, we found that park usage largely returned to 2020 pre-executive order levels (period 2) following the lifting of the shutdown order (period 4). Parks that were closed by the executive order and those that remained open both saw increases in visitation in May (period 4) compared to the park shutdown period (period 3), though parks that were closed by the executive order experienced larger increases in visitation in May than those that had remained open (Fig 5B). Park usage following park reopening was no higher than pre-executive order levels on the whole, however. Visitation in May (period 4) did not differ from visitation in the mid-March/ early April quarantine period before parks were closed (period 2), for both closed and open parks (Fig 5C). Likewise, park usage following the order was still higher than in previous years; park visitation in May, aggregated across all parks, was higher in 2020 as compared to visitation during May of the previous three years (Fig 5A). In fact, of the ten most popular days for park visitation during our study period (2017–2020), seven were in May 2020 (S2A Fig). Thus, the lifting of the shutdown order was associated with increases in park visitation to elevated pre-order levels.

## Discussion

We used social media data as a proxy for park visitation and found that (1) park usage increased substantially (63.4%) with the onset of the pandemic. After the New Jersey governor issued an executive order to close parks, (2) park usage declined substantially and consistently in closed parks, whereas parks that remained open saw variable changes in visitation. Curiously, park usage increased in only a small fraction of open parks (4 of 12) following the executive order, contrary to our predictions (Fig 1). Finally, with park reopening, we found that (3) park usage returned to, but did not exceed, the elevated visitation levels from before the governor's executive order.

### Role of public parks in the pandemic

First and foremost, we observed an increase in public park use with the onset of the COVID-19 pandemic, as measured by posts on social media. This increase indicates that people used public parks more during the beginning of the pandemic for the diverse services that parks provide [1, 6, 23, 49]. Most immediately, people probably visited parks more during the beginning of quarantine due to their accessibility: many public parks are free recreational spaces that are accessible (at least in theory) to diverse communities of people [9–12, 16, 17]. These aspects of parks make them particularly appealing in the face of the widespread financial uncertainty and decreased mobility of quarantine [1, 2, 7, 50]. In addition, with COVID-19-related school closures, many parents were left to engage and educate their children [8]. As such, parks likely served as an outlet for families with children in particular, enabling families to get out of the house and entertain their kids during the pandemic [7, 8, 50, 51]. Finally, parks are known to have positive impacts on mental health (*e.g.*, [9, 17–20].). Therefore, some of the observed increases in park usage during the initial stages of quarantine were likely driven by heightened anxiety and depression as the pandemic became more serious [3–5]. Our findings are consistent with other work demonstrating that people increasingly made use of outdoor spaces for recreation during the pandemic [7, 23]. A study in Oslo, Norway that used mobile phone tracking data found that outdoor recreational activity increased during lockdown [23]. Altogether, such findings emphasize that people increasingly relied on parks during the pandemic for the important and dynamic services that they provide.

### Efficacy of the executive order

We also found that people were highly responsive to the governor's executive order to close parks. We observed substantive declines (-76.1%) in visitation to closed parks associated with the executive order, suggesting that the order was largely effective at deterring park visitation. Likewise, the executive order seemingly did not induce panic and overcrowding; in contrast to our predictions (Fig 1), people did not concentrate in open parks, though a few open parks did see visitation increases. Altogether, this underscores that clear guidance and policies from governing bodies do influence peoples' behavior during moments of crisis [2, 21]. Indeed, this finding parallels other research emphasizing the importance of strong leadership and clear guidelines for handling the pandemic: a recent study found that mask wearing and mask buying both significantly increased following the CDC's formal recommendation that Americans wear face masks [2]. As such, clear leadership and policies are crucial for influencing behavior and managing moments of crisis.

Though the park shutdown order seems to have been largely effective at reducing park usage, this does not mean that it was necessarily the best policy for curtailing the spread of SARS-CoV-2. Other work has suggested that parks are not the transmission centers for SARS-CoV-2 that some have supposed [1, 6, 25]. and may instead even help to mitigate the spread of

the virus [23]. Indeed, we note that reopening parks did not appear to be associated with increases in SARS-CoV-2 cases in New Jersey. In fact, new cases continued to decline even after public parks were reopened in New Jersey and park visitation went back up (Fig 1), though by then additional policies for preventing the spread of SARS-CoV-2 were in place. Still, given such findings and the aforementioned benefits that parks serve, we suggest that park shutdowns may not be the best means of curbing the pandemic. Instead, we should continue to expand our public park infrastructure and increase the accessibility of existing parks so that more people can access the multifaceted benefits that parks provide, particularly in moments of crisis such as those created by the COVID-19 pandemic.

## Lifting of the executive order

While we observed that people largely stopped using parks in response to the park shutdown order, public park usage resumed immediately once parks reopened; once the order was lifted, people visited parks at the same levels that they had been before the order. The immediate return to parks suggests that peoples' relationships to parks are quite plastic, depending on social and recreational context, and that the observed increases in park visitation during the onset of the pandemic were driven largely by the paucity of other recreational options [1, 7, 23, 49]. Likewise, our results suggest that the effects of the executive order were largely reversible; once lifted, the executive order did not cause people to visit parks any more than they had before the order. That visitation quickly returned to pre-restriction levels further emphasizes that people are highly responsive to such orders and policies [26], and also makes clear that policy decisions do not need to be thought of as final or irreversible. Policymakers and leaders can and should explore policy options in real time to evaluate which policies are most effective for dealing with the specific situation at hand. We should not be beholden to the first response we try; there is space to experiment.

## Limitations and future directions

It is possible that other changes in behavior may have contributed to the patterns we observed here, particularly the observed reductions in park visitation to closed parks during the shutdown period. The park shutdown order was instituted during a peak in the number of new SARS-CoV-2 cases in New Jersey (Fig 1). As such, people may have decided to avoid public parks independent of the park shutdown order because they were increasingly nervous to leave home as the situation grew more serious [26, 52]. In addition, observed declines in park visitation with the shutdown order may reflect decreases in Instagram posting rather than declines in park usage itself. People may still have been visiting parks, but not posting photos for fear of social or legal repercussions [53]. To further complicate matters, this effect could be counteracted by retrospective Instagram posts during the shutdown. While we removed all Instagram photos whose captions and tags indicated that they were posted retrospectively, any remaining retrospective posts could have driven some of the observed park usage trends, particularly the usage we observed when parks were closed. However, because such photos would serve to inflate visitation during the shutdown, actual reductions in park usage would be even greater than what we capture here. While we cannot directly evaluate the contribution of these factors to observed patterns, our results suggest an awareness of the executive order regardless; the executive order clearly changed peoples' behaviors, even if not necessarily in the intended way. This result is reassuring, as it means that people clearly knew about the shutdown order. Thus, if the shutdown order had failed, it would not have been for lack of knowledge of the policy. As such, disseminating knowledge of policies such as this is not a major barrier to their effective implementation.

Some have suggested that pandemic restrictions might have ancillary health consequences due to decreased opportunities for recreation and physical activity [50]. Our results suggest that such concerns may be somewhat overexaggerated [21, 53]. Observed increases in the use of outdoor spaces may compensate for overall declines in daily activity levels, thereby enabling people to maintain both their physical and mental well-being [3, 49, 51]. Though we do not have the data to investigate this possibility here, future research should evaluate whether increased use of parks has indeed been able to compensate for other reductions in daily physical activity with lockdown.

While public parks are in theory accessible to everyone, in reality the distribution of publicly available outdoor spaces in New Jersey disproportionately benefits predominantly affluent and predominantly white communities [15, 16]. Communities composed mainly of people of color resultantly have less access to parks and outdoor recreational spaces (and other public infrastructure more generally), despite increased reliance on such spaces for the services they provide [12–16]. A notable aspect of the COVID-19 pandemic in the United States has been its disproportionate impact on these marginalized communities, particularly Black, Hispanic, and Native American communities [54, 55]. In light of our results, we suggest that the lack of access to parks and other public infrastructure may contribute to the disparities in COVID-19 burden observed in these communities, as others have speculated [54, 55]. By having reduced access to parks, these communities are barred from the aforementioned physical and mental health benefits associated with parks at a time when these services are particularly critical [14, 15, 17, 22]. As such, increasing the accessibility and equitability of public infrastructure such as parks may be one avenue for curbing this and future pandemics [27, 54, 55]. To better understand this relationship, further research should focus on how access to public infrastructure relates to SARS-CoV-2 burden, particularly within marginalized communities.

It remains to be seen whether we will see enduring effects of the pandemic on park visitation levels after quarantine is over [6]. Some have speculated that the pandemic will have lasting impacts on how people interact with public spaces, causing people to both avoid certain public spaces and aggregate in others [6, 49], but our findings suggest that this may not be the case. We observed that people returned to parks quite quickly once they reopened, and that park visitation rates were comparable to those before the shutdown. Therefore, once restrictions are lifted, many people will likely go back to their alternative forms of recreation [6, 49], and our results suggest that these behavioral changes could be quite immediate [3, 6, 22]. Overall, our findings serve to underscore the adaptability and resilience that people have exhibited during the COVID-19 pandemic [3, 6].

## Conclusions

Our study has two clear implications. First, we found that public parks continue to serve a crucial role as recreational spaces and health resources during the COVID-19 pandemic [9–12, 17], particularly in circumstances where recreational opportunities are limited [6, 7, 22, 23]. Throughout the world, and particularly in New Jersey, the human population is expanding [34, 36, 56], meaning that the demand for parks and outdoor recreational spaces is likely to increase. Given their well-documented benefits and ongoing functionality, we suggest that legislators and policymakers should continue to invest in public parks and expand access so as to ensure that they are available to diverse communities.

Likewise, our study has more general implications for introducing novel policies in times of crisis to influence human behavior. We find that such policies do have a notable effect on how people behave and therefore can be an effective means of mitigating crises like the COVID-19 pandemic [2, 21]. Still, our results underscore that there is space for exploration; policies are

not final and can be reversed with little adverse effect. We should therefore continue to enact exploratory policies for regulating public space use and consider diverse, creative options for dealing with crises such as the COVID-19 pandemic. As extreme situations become more frequent with expanding global populations and accelerating land use conversion [56], such an understanding of how people react to policy decisions will be crucial for planning our responses to future crises.

## Supporting information

**S1 Fig. Changes in visitation (as measured by PUD) in the four open parks where visitation increased after the governor's executive order.** Visitation to Schiff Nature Preserve almost doubled, whereas increases were much more moderate in Bear Creek Preserve, Black River Wildlife Management Area, and Plainsboro Preserve. The numbers in red circles refer to the time periods denoted in Fig 1.
(TIF)

**S2 Fig.** (a) Total park visitation (PUD summed across all parks) during the study period (2017–2020) as well as (b) average visitation for parks that were closed or open during the executive order. The black line in (a) reflects the trend in the number of active US Instagram users through time, which was used to detrend PUD values when comparing park usage across years. Note in (a) that Instagram usage has increased through time, and also the distinct seasonality of park visitation. Likewise, note in (b) that parks that were closed during the park shutdown consistently had higher visitation historically than parks that remained open.
(TIF)

**S1 File. List of sources consulted to determine park closure status during the shutdown.**
(DOCX)

## Author Contributions

**Conceptualization:** Zoe M. Volenec, Andy P. Dobson.

**Data curation:** Zoe M. Volenec, Joel O. Abraham.

**Formal analysis:** Zoe M. Volenec, Joel O. Abraham, Alexander D. Becker.

**Investigation:** Zoe M. Volenec, Joel O. Abraham.

**Methodology:** Zoe M. Volenec, Joel O. Abraham, Alexander D. Becker.

**Project administration:** Joel O. Abraham.

**Supervision:** Andy P. Dobson.

**Visualization:** Zoe M. Volenec, Joel O. Abraham.

**Writing – original draft:** Joel O. Abraham.

**Writing – review & editing:** Zoe M. Volenec, Joel O. Abraham, Alexander D. Becker, Andy P. Dobson.

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
