## [Decision Letter · Decision Letter 0]

25 Feb 2021

PONE-D-20-35168

Public parks and the pandemic: how park usage has been affected by COVID-19 policies

PLOS ONE

Dear Dr. Abraham,

Thank you for submitting your manuscript to PLOS ONE. After careful consideration, we feel that it has merit but does not fully meet PLOS ONE’s publication criteria as it currently stands. Therefore, we invite you to submit a revised version of the manuscript that addresses the points raised during the review process that are noted below.

We look forward to receiving your revised manuscript.

Kind regards,

Christopher A. Lepczyk

Academic Editor

PLOS ONE

Journal Requirements:

3. We note that Figure 2 in your submission contains map images which may be copyrighted. All PLOS content is published under the Creative Commons Attribution License (CC BY 4.0), which means that the manuscript, images, and Supporting Information files will be freely available online, and any third party is permitted to access, download, copy, distribute, and use these materials in any way, even commercially, with proper attribution. For these reasons, we cannot publish previously copyrighted maps or satellite images created using proprietary data, such as Google software (Google Maps, Street View, and Earth). For more information, see our copyright guidelines: http://journals.plos.org/plosone/s/licenses-and-copyright.

(1) You may seek permission from the original copyright holder of Figure(s) [#] to publish the content specifically under the CC BY 4.0 license. 

(3) Please could you provide a statement indicating that the use of this dataset was done in compliance to the Instagram Terms and Conditions and our requirements for this type of study (https://journals.plos.org/plosone/s/submission-guidelines#loc-personal-data-from-third-party-sources).

Thanks for your attention to our requests.

Additional Editor Comments:

Subject Editor Comments:

Overall this is a nicely written and well done manuscript. Both the reviewers and I have only a moderate number of comments.

L30. Please add in a few sentences of methods before moving to results.

L30. Suggest giving some specific values here, such as percent increase from before pandemic.

L32. Again, by what percent or volume? Providing some measures of change in Abstract would be very helpful.

L41. Delete ‘ongoing’ so the paper holds relevance years from now when it is no longer ongoing.

L91. I would suggest you have a sentence here that states your main goal of the research, then follow it with the research questions and sentences.

L91-97. Move these to Methods.

L93. ‘These data.’

L107. Please move Methods here. As reviewers note it would be helpful to have them here as the journal isn’t stylized like Science or Nature.

L109. Change ‘hypothesized’ to ‘predicted’ as that is how you stated in Introduction.

L125. Do you mean ‘On the whole?’

L157. Change ‘relied’ to ‘used’ or similar as you did not measure reasons why people are in parks.

L179-86. You do not have data to support these statements, please cut this paragraph.

L247. But how would you have tested or demonstrated that relationship? Seems somewhat difficult to test without data on individuals.

L261, L275, L284. What does ‘this’ refer to at beginning of sentence? Need object after ‘this’ to logically connect to previous sentence. Please revise throughout ms.

L293-97, L303-306. These are important points that people will agree with, but they are not really about your data or results. You are simply looking at visitation rates before, during, and after parks were closed and you can’t really go much beyond that for what you found as you are not measuring people or any social item.

L312. Approximately what percent of parks in the state does this represent?

L342. What is the spatial error you accepted for a tag to be within a park? That is, there are likely tags that are on park boundaries that could legitimately be inside or outside based on error. How did you account for this error?

Figure 1. In this figure you present hypotheses, but in legend and text you are discussing predictions. These are different and thus need to make figures and text consistent with what you mean.

Reviewer 1. Note that the PDF copy has mark up on it, but may be difficult to see all of the comments. Thus, they are listed below as well:

Dear authors, I commend your work during this unique and challenging period in history. I think this study is important, and adds to the growing body of research using social media data to improve understanding of visitor use-descriptive data to understand behavioral patterns. This is unique in the sense that it monitors shifts during the COVID pandemic, and specifically park use during this time.

I recognize that the format of this journal places the emphasize on the methodological approach toward the end of the manuscript, rather than following the Introduction. However, I believe there are pertinent details that should be included earlier in the manuscript to help validate the study. For example, more detail should be provided regarding the types of parks included in the study, specifically highlighting agency, designation, and in general, governing body, as regardless of state mandates, agencies operated differently during closures, and this inherently influenced the results of this study.

Additional details should be provided regarding Instagram usage (e.g., percentage of NJ residents that use Instagram; percentage in use per the years compared in this study; representativeness of this study compared to the general populous, etc.). This also leads to a need for a clear limitations section, which I believe merits a separate header/section, which could directly proceed future research recommendations. Even though the methods come at the end of the document, figures should follow the flow of the paper, and therefore need adjusted as the paper is revised.

Finally, in several areas within the Results and Discussion, there are topics of importance that are raised, that are more speculative rather than grounded in the results of this paper. For example, issues with parks, equity, and access are certainly important, and well-studied, and could even likely be evaluated using your dataset, GIS, and socioeconomic data for the areas of study, but this study does not directly address that issue. The areas of speculation should be grounded in existing research, and stated as future research directions stemming from this work.

The above comments are the most pressing from my perspective, but there were a few other minor considerations that should be addressed. The paper is well-written, and very easy to follow, but it does need additional literature in several areas, specific to both COVID and recreation (examples added in the commented paper copy), and park use, associated benefits, and equity. While the COVID and recreation literature is rapidly changing, the other topics need more attention with research from the past decade and beyond. Finally, other literature should be added highlighting the use of social media data for monitoring park use. There have been many studies looking at Twitter, Instagram and other outlets published in the past decade. As for the storyline, there is some discussion about the use of parks for parents and children, but that is a bit lost in the results. I’m not sure that direction is additive to the introduction as currently stated. Overall, great work, and I believe this manuscript merits publication after some revisions.

Line 28: Clarity should be given regarding types of parks. State? Local? etc.

Line 55: While this is a bit of a moving target as empirical works continue to be published on recreation and covid, I would add a few other published citations here:

http://doi.org/10.1093/jue/juaa020

https://outdoorindustry.org/article/increase-outdoor-activities-due-covid-19/

Line 84: Thus far the manuscript has been discussing what seems like playground-types of local parks. This mandate was specific to state and county parks. Clarity at the beginning and throughout the paper regarding what type of parks are being discussed is important.

Line 91: Is this the Methods section? This needs to be identified and more information needs to be included to highlight the methodology applied to this study

Line 101: This is written in an exploratory manner at the beginning, but here anticipated findings are suggested (not quite hypotheses just yet), but Figure 1 is presented as hypotheses. I think the paper would be strengthened by keep it language as more exploratory rather than as hypotheses. If the authors opt for using hypotheses, properly state them as such (i.e., less exploratory).

Line 108: Figure 2 seems to be skipped in the text. But, Figure 2 needs much more finite info rather than the two dichotomous choices presented as parks closed or not closed.

Line 116: Was there a difference in designation regarding what was closed versus open? For example, how were closed parks determined?

Line 158: The first few citations here don’t really represent this sentence (Ulrich; Godbey; Manning).

Line 170: This is a big body of literature so I recommend an “e.g.” for these citations)

Line 179: I believe a header specifically highlighting this section as “future research” or similar is needed. Some of this is speculative, but I don’t disagree with any of these suggestions. But, I believe it would be better suited if presented with future research framing.

Line 219: Seems incorrect grammatically

Line 223: A limitations section is needed. Some of the content here would be included.

Line 232: Need clarity on where this section resides

Line 291: They are health resources and should be stated as such, throughout this paper. See Andrew Mowen and colleagues works for more citations

Reviewers' comments:

Reviewer's Responses to Questions

**Comments to the Author**

1. Is the manuscript technically sound, and do the data support the conclusions?

Reviewer #1: Yes

Reviewer #2: Yes

2. Has the statistical analysis been performed appropriately and rigorously? 

Reviewer #1: Yes

Reviewer #2: Yes

3. Have the authors made all data underlying the findings in their manuscript fully available?

Reviewer #1: Yes

Reviewer #2: Yes

4. Is the manuscript presented in an intelligible fashion and written in standard English?

Reviewer #1: Yes

Reviewer #2: Yes

5. Review Comments to the Author

Reviewer #1: Dear authors, I commend your work during this unique and challenging period in history. I think this study is important, and adds to the growing body of research using social media data to improve understanding of visitor use-descriptive data to understand behavioral patterns. This is unique in the sense that it monitors shifts during the COVID pandemic, and specifically park use during this time.

I recognize that the format of this journal places the emphasize on the methodological approach toward the end of the manuscript, rather than following the Introduction. However, I believe there are pertinent details that should be included earlier in the manuscript to help validate the study. For example, more detail should be provided regarding the types of parks included in the study, specifically highlighting agency, designation, and in general, governing body, as regardless of state mandates, agencies operated differently during closures, and this inherently influenced the results of this study.

Additional details should be provided regarding Instagram usage (e.g., percentage of NJ residents that use Instagram; percentage in use per the years compared in this study; representativeness of this study compared to the general populous, etc.). This also leads to a need for a clear limitations section, which I believe merits a separate header/section, which could directly proceed future research recommendations. Even though the methods come at the end of the document, figures should follow the flow of the paper, and therefore need adjusted as the paper is revised.

Finally, in several areas within the Results and Discussion, there are topics of importance that are raised, that are more speculative rather than grounded in the results of this paper. For example, issues with parks, equity, and access are certainly important, and well-studied, and could even likely be evaluated using your dataset, GIS, and socioeconomic data for the areas of study, but this study does not directly address that issue. The areas of speculation should be grounded in existing research, and stated as future research directions stemming from this work.

The above comments are the most pressing from my perspective, but there were a few other minor considerations that should be addressed. The paper is well-written, and very easy to follow, but it does need additional literature in several areas, specific to both COVID and recreation (examples added in the commented paper copy), and park use, associated benefits, and equity. While the COVID and recreation literature is rapidly changing, the other topics need more attention with research from the past decade and beyond. Finally, other literature should be added highlighting the use of social media data for monitoring park use. There have been many studies looking at Twitter, Instagram and other outlets published in the past decade. As for the storyline, there is some discussion about the use of parks for parents and children, but that is a bit lost in the results. I’m not sure that direction is additive to the introduction as currently stated. Overall, great work, and I believe this manuscript merits publication after some revisions.

Reviewer #2: Great job here - I think this will be a good paper for understanding anecdotal accounts from managers with empirical data.

My one gripe: these social media data are great, but they are not a panacea. You should do a better job of spending at least a paragraph - and preferably more - about the limitations and biases of your data.

6. PLOS authors have the option to publish the peer review history of their article (what does this mean?). If published, this will include your full peer review and any attached files.

Reviewer #1: No

Reviewer #2: No

---

## [Author Response · Author response to Decision Letter 0]

11 Apr 2021

Editor Comments:

We note that Figure 2 in your submission contains map images which may be copyrighted. All PLOS content is published under the Creative Commons Attribution License (CC BY 4.0), which means that the manuscript, images, and Supporting Information files will be freely available online, and any third party is permitted to access, download, copy, distribute, and use these materials in any way, even commercially, with proper attribution. For these reasons, we cannot publish previously copyrighted maps or satellite images created using proprietary data, such as Google software (Google Maps, Street View, and Earth). For more information, see our copyright guidelines: http://journals.plos.org/plosone/s/licenses-and-copyright. We require you to either (1) present written permission from the copyright holder to publish these figures specifically under the CC BY 4.0 license, or (2) remove the figures from your submission.

We have revised Figure 2 so that it no longer includes copyrighted material.

Please could you provide a statement indicating that the use of this dataset was done in compliance to the Instagram Terms and Conditions and our requirements for this type of study (https://journals.plos.org/plosone/s/submission-guidelines#loc-personal-data-from-third-party-sources).

We have added such a statement both in the text, to line 148, and also in the data accessibility statement.

Subject Editor Comments:

Overall this is a nicely written and well done manuscript. Both the reviewers and I have only a moderate number of comments.

We thank the editor for their careful consideration of our manuscript, and for the reviewers’ thoughtful comments regarding how to improve the clarity of the paper.

L30. Please add in a few sentences of methods before moving to results.

We have added some more detail about the study methodology here, as suggested.

L30. Suggest giving some specific values here, such as percent increase from before pandemic.

We have added the magnitude of the increase in park usage here as well as throughout the text.

L32. Again, by what percent or volume? Providing some measures of change in Abstract would be very helpful.

 We have added the magnitude of the decrease in park usage here as well as throughout the text.

L41. Delete ‘ongoing’ so the paper holds relevance years from now when it is no longer ongoing.

OK.

L91. I would suggest you have a sentence here that states your main goal of the research, then follow it with the research questions and sentences.

Noted, and revised accordingly.

L91-97. Move these to Methods.

We have moved these sentences to the beginning of the Methods, as suggested.

L93. ‘These data.’

Thank you for pointing out this grammatical error, we have fixed it.

L107. Please move Methods here. As reviewers note it would be helpful to have them here as the journal isn’t stylized like Science or Nature.

Noted, and revised accordingly.

L109. Change ‘hypothesized’ to ‘predicted’ as that is how you stated in Introduction.

OK.

L125. Do you mean ‘On the whole?’

Yes, thank you for pointing this out.

L157. Change ‘relied’ to ‘used’ or similar as you did not measure reasons why people are in parks.

Noted, and revised accordingly.

L179-86. You do not have data to support these statements, please cut this paragraph.

Per the reviewer suggestions, we have added a ‘Limitations and future directions’ section to our discussion. We have removed this paragraph but have moved some of the content there.

L247. But how would you have tested or demonstrated that relationship? Seems somewhat difficult to test without data on individuals.

As you said, we did not have the data to explicitly test this here. We intended this to convey that we were merely noting the lack of an uptick in cases with park reopening. We have revised the language in this section to convey this fact more clearly.

L261, L275, L284. What does ‘this’ refer to at beginning of sentence? Need object after ‘this’ to logically connect to previous sentence. Please revise throughout ms.

Thank you for making us aware of this issue. We have revised for this source of ambiguity throughout the manuscript.

L293-97, L303-306. These are important points that people will agree with, but they are not really about your data or results. You are simply looking at visitation rates before, during, and after parks were closed and you can’t really go much beyond that for what you found as you are not measuring people or any social item.

While we agree that these points are not the direct findings of the current work, we feel these ideas follow logically from our results and are important takeaways of this study. In response to this comment, we have somewhat revised the language in this section, but we would like to keep this material in the paper if possible. 

L312. Approximately what percent of parks in the state does this represent?

Unfortunately, the data do not exist to adequately address this question. Statewide geodatabases of open spaces suggest that there are approximately 5,500 uniquely identifiable open spaces in the state of New Jersey, which would mean that our study captures ~1.80% of the open spaces within the state. However, such databases are not very accurate (parks are often duplicated or misidentified), nor are they updated consistently, so we are hesitant to include this statistic in the paper. 

L342. What is the spatial error you accepted for a tag to be within a park? That is, there are likely tags that are on park boundaries that could legitimately be inside or outside based on error. How did you account for this error?

This comment reflects some misunderstanding about how Instagram geotags operate. Instagram users are able to self-select the geotags that correspond to their photograph from a list of possible geotags within the application. As such, there is no means by which to explicitly evaluate this kind of error. We have attempted to clarify this somewhat on line 144.

Figure 1. In this figure you present hypotheses, but in legend and text you are discussing predictions. These are different and thus need to make figures and text consistent with what you mean.

Noted, and revised accordingly.

Reviewer #1: 

Note that the PDF copy has mark up on it, but may be difficult to see all of the comments. Thus, they are listed below as well:

Dear authors, I commend your work during this unique and challenging period in history. I think this study is important, and adds to the growing body of research using social media data to improve understanding of visitor use-descriptive data to understand behavioral patterns. This is unique in the sense that it monitors shifts during the COVID pandemic, and specifically park use during this time.

We thank you for your kind comments about the novel aspects of this work and thoughtful suggestions for the improvement of the manuscript.

I recognize that the format of this journal places the emphasize on the methodological approach toward the end of the manuscript, rather than following the Introduction. However, I believe there are pertinent details that should be included earlier in the manuscript to help validate the study. For example, more detail should be provided regarding the types of parks included in the study, specifically highlighting agency, designation, and in general, governing body, as regardless of state mandates, agencies operated differently during closures, and this inherently influenced the results of this study.

We entirely agree with this comment. The editor has suggested that we move our Methods and materials section to follow the Introduction, which we have done. We feel this adds clarity about our approach and provides the information earlier on in the paper that you have suggested adding.

Additional details should be provided regarding Instagram usage (e.g., percentage of NJ residents that use Instagram; percentage in use per the years compared in this study; representativeness of this study compared to the general populous, etc.). 

Unfortunately, information about Instagram usage by state is not publicly available. As such, we are not able to address this here. However, considerable work has gone into validating that studies relying on social media data are representative of population-level behavior (see Wood et al. 2013, Donahue et al. 2017, and Hamstead et al. 2018, which we cite throughout the text).

This also leads to a need for a clear limitations section, which I believe merits a separate header/section, which could directly proceed future research recommendations. 

Per this comment and similar suggestions from both the editor and the other reviewer, we have added a ‘Limitations and future directions’ section to our discussion.

Even though the methods come at the end of the document, figures should follow the flow of the paper, and therefore need adjusted as the paper is revised.

We feel the order of the figures is now logical given the revised structure of the paper.

Finally, in several areas within the Results and Discussion, there are topics of importance that are raised, that are more speculative rather than grounded in the results of this paper. For example, issues with parks, equity, and access are certainly important, and well-studied, and could even likely be evaluated using your dataset, GIS, and socioeconomic data for the areas of study, but this study does not directly address that issue. The areas of speculation should be grounded in existing research, and stated as future research directions stemming from this work.

We are glad to hear that you feel these sections have merit; we have moved these more speculative portions of the discussion to the ‘Limitations and future directions’ section that we have added in response to your comments and those of the other reviewer and subject editor.

The above comments are the most pressing from my perspective, but there were a few other minor considerations that should be addressed. The paper is well-written, and very easy to follow, but it does need additional literature in several areas, specific to both COVID and recreation (examples added in the commented paper copy), and park use, associated benefits, and equity. While the COVID and recreation literature is rapidly changing, the other topics need more attention with research from the past decade and beyond. Finally, other literature should be added highlighting the use of social media data for monitoring park use. There have been many studies looking at Twitter, Instagram and other outlets published in the past decade. 

Thank you for your suggestions below regarding other literature to reference. We have added these citations as well as several other citations from both the recreation literature and evolving COVID-19 literature.

As for the storyline, there is some discussion about the use of parks for parents and children, but that is a bit lost in the results. I’m not sure that direction is additive to the introduction as currently stated. 

We agree that this was overemphasized in the original draft and have removed this sentence from the Introduction.

Overall, great work, and I believe this manuscript merits publication after some revisions.

Thank you!

Line 28: Clarity should be given regarding types of parks. State? Local? etc.

We have added this information here and have further clarified the types of park included in this study on lines 28-29.

Line 55: While this is a bit of a moving target as empirical works continue to be published on recreation and covid, I would add a few other published citations here:

http://doi.org/10.1093/jue/juaa020

https://outdoorindustry.org/article/increase-outdoor-activities-due-covid-19/

Thank you for pointing us to these sources, we have cited them here and throughout the paper.

Line 84: Thus far the manuscript has been discussing what seems like playground-types of local parks. This mandate was specific to state and county parks. Clarity at the beginning and throughout the paper regarding what type of parks are being discussed is important.

We have added some clarity in the Methods about how we defined parks for this study (see lines 131-132). As such, we hope that the restructuring of the paper adequately addresses this comment. 

Line 91: Is this the Methods section? This needs to be identified and more information needs to be included to highlight the methodology applied to this study.

We have moved this information to the beginning of the Methods section. Also, we have relocated the Methods section so it now comes after the Introduction, per standard format.

Line 101: This is written in an exploratory manner at the beginning, but here anticipated findings are suggested (not quite hypotheses just yet), but Figure 1 is presented as hypotheses. I think the paper would be strengthened by keep it language as more exploratory rather than as hypotheses. If the authors opt for using hypotheses, properly state them as such (i.e., less exploratory).

We have revised Figure 1 to say ‘predictions’ rather than ‘hypotheses’.

Line 108: Figure 2 seems to be skipped in the text. But, Figure 2 needs much more finite info rather than the two dichotomous choices presented as parks closed or not closed.

See our response to the above comment regarding figure ordering. We initially referred to Figure 2 on line 92. Now that the Methods section has been relocated and this content has been moved to the beginning of the Methods, we first refer to Figure 2 on line 119. Also, Figure 2 has been largely revised.

Line 116: Was there a difference in designation regarding what was closed versus open? For example, how were closed parks determined?

We discussed this in the Methods on lines 213-221. Hopefully this is clearer from the outset now that we have moved the Methods section up.

Line 158: The first few citations here don’t really represent this sentence (Ulrich; Godbey; Manning).

Noted, and revised accordingly.

Line 170: This is a big body of literature so I recommend an “e.g.” for these citations)

We have added ‘e.g.’, as suggested.

Line 179: I believe a header specifically highlighting this section as “future research” or similar is needed. Some of this is speculative, but I don’t disagree with any of these suggestions. But, I believe it would be better suited if presented with future research framing.

Per this suggestion, we have added a ‘Limitations and future directions’ section at the end of the discussion.

Line 219: Seems incorrect grammatically

We agree that this sentence was confusing and have revised it accordingly.

Line 223: A limitations section is needed. Some of the content here would be included.

See above comment; we have added a ‘Limitations and future directions’ section at the end of the discussion.

Line 232: Need clarity on where this section resides

We hope that relocating the Methods section has addressed this concern.

Line 291: They are health resources and should be stated as such, throughout this paper. See Andrew Mowen and colleagues works for more citations

Noted, and revised accordingly.

Reviewer #2: 

Great job here - I think this will be a good paper for understanding anecdotal accounts from managers with empirical data.

Thank you for these kind comments!

My one gripe: these social media data are great, but they are not a panacea. You should do a better job of spending at least a paragraph - and preferably more - about the limitations and biases of your data.

Noted. We agree that social media data have their limitations, and as such we have added a ‘Limitations and future directions’ section at the end of the discussion to be more explicit about this. Also, we discuss the limitations of using social media data quite extensively in the Methods section (lines 174-186), which we have relocated.

---

## [Editor Report · Decision Letter 1]

4 May 2021

Public parks and the pandemic: how park usage has been affected by COVID-19 policies

PONE-D-20-35168R1

Dear Dr. Abraham,

We’re pleased to inform you that your manuscript has been judged scientifically suitable for publication and will be formally accepted for publication once it meets all outstanding technical requirements.

Kind regards,

Christopher A. Lepczyk

Academic Editor

PLOS ONE
---

## [Editor Report · Acceptance letter]

7 May 2021

PONE-D-20-35168R1 

Public parks and the pandemic: how park usage has been affected by COVID-19 policies 

Dear Dr. Abraham:

I'm pleased to inform you that your manuscript has been deemed suitable for publication in PLOS ONE. Congratulations! Your manuscript is now with our production department. 

Kind regards, 

on behalf of

Dr. Christopher A. Lepczyk 

Academic Editor

PLOS ONE